:ᗺ: PLOS | ONE

# Enhancing performance of subject-specific models via subject-independent information for SSVEP-based BCIs

**Mohammad Hadi Mehdizavareh**[1], **Sobhan Hemati**[1], **Hamid Soltanian-Zadeh**[1,2]*

**1** CIPCE, School of Electrical and Computer Engineering, College of Engineering, University of Tehran, Tehran, Iran, **2** Medical Image Analysis Laboratory, Departments of Radiology and Research Administration, Henry Ford Health System, Detroit, MI, United States of America

* hszadeh@ut.ac.ir, hsoltan1@hfhs.org

**Data Availability Statement:** The data that support the findings of this study are openly available at "ftp://sccn.ucsd.edu/pub/ssvep_benchmark_dataset/. Further questions can be directed to the

## Abstract

Recently, brain-computer interface (BCI) systems developed based on steady-state visual evoked potential (SSVEP) have attracted much attention due to their high information transfer rate (ITR) and increasing number of targets. However, SSVEP-based methods can be improved in terms of their accuracy and target detection time. We propose a new method based on canonical correlation analysis (CCA) to integrate subject-specific models and subject-independent information and enhance BCI performance. We propose to use training data of other subjects to optimize hyperparameters for CCA-based model of a specific subject. An ensemble version of the proposed method is also developed for a fair comparison with ensemble task-related component analysis (TRCA). The proposed method is compared with TRCA and extended CCA methods. A publicly available, 35-subject SSVEP benchmark dataset is used for comparison studies and performance is quantified by classification accuracy and ITR. The ITR of the proposed method is higher than those of TRCA and extended CCA. The proposed method outperforms extended CCA in all conditions and TRCA for time windows greater than 0.3 s. The proposed method also outperforms TRCA when there are limited training blocks and electrodes. This study illustrates that adding subject-independent information to subject-specific models can improve performance of SSVEP-based BCIs.

## 1. Introduction

Brain-computer interface (BCI) systems provide novel communication channels for the humans, especially severely disabled individuals [1–3]. A character speller system is a highly important BCI system which allows disabled individuals to communicate with their surrounding environment [2]. Electroencephalography (EEG) is a noninvasive, low cost, and simple modality, widely used to implement BCI spellers [4]. In recent years, steady-state visual evoked potential (SSVEP)-based BCI spellers have attracted much more attention compared with other BCI systems including motor imagery and P300. This is because of their high information transfer rate (ITR), less user training, and ability to deal with problems with a large number of classes [4–7].

data owners here: Yijun Wang: wangyj@semi.ac.cn
Xiaogang Chen: chenxg@bme.cams.cn Xiaorong
Gao: gxrdea@tsinghua.edu.cn Shangkai Gao: gsk-
dea@tsinghua.edu.cn

**Funding:** The authors received no specific funding
for this work.

**Competing interests:** The authors have declared
that no competing interests exist.

There are many target coding methods in SSVEP-based BCIs, among which frequency coding is a popular method to encode targets [8, 9]. Several methods have been proposed to combine phase and frequency coding approaches [10–12]. The most discriminative method is joint frequency-phase modulation (JFPM) method which assigns different frequencies and phases to two adjacent targets [12]. Target identification is another crucial issue in SSVEP-based BCIs, for which numerous methods have been proposed. Initially, single-channel methods were presented based on power spectral density analysis (PDSA) [13–14] and then multiple channel methods were introduced to improve the signal-to-noise ratio (SNR) of the SSVEP response. In these methods, channels are combined using appropriate spatial filters so that common noises in the channels are reduced and the quality of SSVEP response is improved. Some powerful examples of such methods are minimum energy combination (MEC) [15], maximum contrast combination (MCC) [15], and canonical correlation analysis (CCA) [16]. Although these methods are widely used because of simplicity and no need for training, they only detect frequency. They are unable to discriminate two different phases [11] and their performance degrades in short time windows due to background noise of EEG. To solve these problems, calibration data has been used [12, 17–20].

Extended CCA was introduced to combine CCA coefficients with the Pearson correlation coefficients of the test and training data [12]. Multiway CCA (MwayCCA) [17], L1-regularized MwayCCA [18], and multiset CCA (MsetCCA) [19] were proposed to optimize artificial sine-cosine reference signals embedded in CCA using training trials of each subject. Also, task-related component analysis (TRCA) was suggested to enhance the SNR of the SSVEP response using optimized spatial filters [20]. TRCA extracts task-related components by maximizing the reproducibility during the task period [21]. Comparison studies have shown that extended CCA and TRCA methods are superior to other methods in terms of classification accuracy and ITR, especially in short time windows [20, 22]. Thus, we compare our proposed method with these two methods.

From training point of view, target identification methods can be classified into three main categories [23]: 1) training-free methods such as PSDA and CCA, which do not need any calibration data; 2) subject-specific training methods such as extended CCA and TRCA, for which calibration data are collected for each subject and the parameters of the algorithm are optimized individually; and 3) subject-independent training methods like transfer template-based CCA (tt-CCA) [24], which use the training data of the existing subjects to create a fixed model for a new subject.

In this paper, we propose a new CCA-based method which exploits both subject-specific and subject-independent training methods to enhance performance of a BCI system. A publicly available, 35-subject SSVEP benchmark dataset [25] is used to evaluate the proposed method. First, the most informative CCA-based correlation coefficients are found using a subject-independent training method and then, the selected coefficients are used for a new subject. Also, an ensemble version of the CCA-based method is introduced in which a linear combination of the correlation coefficients derived from the basic and ensemble spatial filters are used to construct the final feature for target identification.

The remainder of the paper is organized as follows. Section 2 introduces benchmark dataset and data preprocessing applied to all methods and reviews standard CCA, extended CCA, and TRCA methods. Then, the basic and ensemble version of the proposed algorithm is described in details, and finally, filter bank analysis is explained. Section 3 presents the experimental results. In section 4, the difference between the proposed algorithm and the extended CCA method is discussed, and the advantages of our method over other methods are described. Section 5 concludes the paper.

## 2. Methods

### 2.1. Benchmark dataset

In this study, the benchmark dataset introduced in [25] has been used. This dataset is freely available to the BCI community to facilitate comparison of the SSVEP response detection algorithms. The dataset has been collected from 35 subjects (17 females, 18 males, a mean age of 22 years, 27 naïve, and 8 experienced). The experiment includes a 40-target speller system which uses the JFPM method to encode characters with 0.2 Hz frequency difference and $0.5\pi$ phase difference between the two neighboring targets. Also, the frequency interval used in this task is in the range of [8, 15.8] Hz. It has been shown that the phase interval of $0.35\pi$ leads to the best performance of the BCI system [12]. Thus, the method proposed in [12, 25] is used to shift the EEG data circularly such that the phase difference is $0.35\pi$. For each subject, the task consists of six blocks and each block includes 40 trials, one trial for each target randomly presented through the LCD to the subjects. In each trial, a visual cue (red square) is shown on the screen for 0.5 s and the subjects are asked to follow the cue target on the screen using their eyes. As the cue disappears, all 40 targets start flickering simultaneously for 5 s. When the stimuli is finished, the screen becomes blank for 0.5 s before the next trial starts. Therefore, each trial lasts 6 s. In every block, the subjects are asked to avoid blinking during stimulus presentation. To avoid eye fatigue, there are several minutes of rest between the two successive blocks.

The EEG data were acquired from 64 channels using Synamps2 system (Neuroscan Company) with a sampling rate of 1000 Hz. The electrodes were placed according to the international 10–20 system. The ground electrode was placed between Fz and FPz and the reference electrode was placed at the vertex. The passband of the amplifier was between 0.15 Hz and 200 Hz, and the electrode impedances were kept less than 10 kΩ. Also, during data recording, a notch filter was used to remove the 50 Hz power line noise. The synchronous signal generated by the stimulus program was sent to the amplifier and recorded on an event channel synchronized to the visual cue onset. To reduce the data size, all EEG epochs were down-sampled to 250 Hz. Further details of the dataset are given in [25].

### 2.2. Data preprocessing

The first step of the EEG data preprocessing is channel selection. The SSVEP topographic scalp maps show high activity over the parietal and visual areas [26, 27]. Based on the previous studies [12, 25], nine electrodes located in these areas (O1, O2, Oz, PO3, PO4, PO5, PO6, POz, and Pz) are selected. By taking into account the 140 ms latency of the visual system [12, 28], for a time window with length Tw s, all epochs are extracted in the interval [0.14 s 0.14+Tw s] in which the time 0 indicates the stimulus onset. Then, all segmented epochs are band-pass filtered from 6 Hz to 90 Hz using a zero-phase Chebyshev Type II infinite impulse response (IIR) filter. The filtfilt() function in MATLAB is used to implement zero-phase forward and reverse filtering.

### 2.3. Reference methods

**2.3.1. Standard CCA method.** CCA is a statistical multivariate method to maximize the correlation between two sets of variables and has been widely used in SSVEP-based BCI for frequency detection [16, 29]. Let $f_K$, $F_s$, $N_t$, $M$, $K$, and $N_h$ denote the $k$-th stimulus frequency, the sampling rate, the number of time points, the EEG channels, the targets, and the harmonic frequencies considered, respectively. The multichannel EEG data is represented by $\mathbf{X} \in \mathbb{R}^{M \times N_t}$

and the reference signals $\mathbf{Y}_k \in \mathbb{R}^{2N_h \times N_t}$ are sinusoidal and defined as:

$$\mathbf{Y}_k = [\mathbf{y}(t_1)\mathbf{y}(t_2)\dots\mathbf{y}(t_{N_t})],$$

$$\mathbf{y}(t) = \begin{pmatrix} \sin(2\pi f_K t) \\ \cos(2\pi f_K t) \\ \vdots \\ \sin(2\pi N_h f_K t) \\ \cos(2\pi N_h f_K t) \end{pmatrix}, \ t = \frac{1}{F_s}, \frac{2}{F_s}, \dots, \frac{N_t}{F_s} \tag{1}$$

CCA finds the weight vectors $w_x$ and $w_y$ so that the correlation between two canonical variables $x = \mathbf{X}^T w_x$ and $y = \mathbf{Y}_k^T w_y$ (which are linear combinations of $\mathbf{X}$ and $\mathbf{Y}_k$ respectively) is maximized by solving the following optimization problem [16]:

$$\rho_k = \max_{w_x, w_y} \rho(x, y) = \frac{E[x^T y]}{\sqrt{E[x^T x]E[y^T y]}} = \frac{E[w_x^T \mathbf{X}\mathbf{Y}_k^T w_y]}{\sqrt{E[w_x^T \mathbf{X}\mathbf{X}^T w_x]E[w_y^T \mathbf{Y}_k \mathbf{Y}_k^T w_y]}} \tag{2}$$

where $\rho(x,y)$ is the Pearson's correlation coefficient between $x$ and $y$ and $\rho_k$ is the maximum of $\rho$ with respect to $w_x$ and $w_y$. To recognize the frequency of SSVEP, $\rho_k$ is calculated for all targets ($k = 1,2,\dots,K$) and the target with the maximal $\rho_k$ is selected as:

$$k^* = \arg\max_k \rho_k, \quad k = 1, 2, \dots, K \tag{3}$$

**2.3.2. Extended CCA-based method.** The standard CCA method is an unsupervised method, meaning that it does not use any calibration data for target identification. This method has been originally developed for frequency detection. Since phase detection requires training data, CCA cannot be used to distinguish different phases [7]. Incorporating training data in target identification methods can capture the temporal features of SSVEP response more effectively and enhance the performance of the CCA-based approaches [12, 22]. Extended CCA which combines standard CCA and individual training-based methods has been proposed in several studies [5, 7, 12, 30] and its superiority over other CCA-based training methods has been shown in [22]. In this method, individual SSVEP template signals $\hat{\mathbf{X}}_k$ are derived by averaging multiple training trials related to the $k$-th target. Then, projections of a test data $\mathbf{X}$ and an individual template $\hat{\mathbf{X}}_k$ are computed using the CCA-based spatial filters, and finally, the correlation coefficients between some pairs of the projections are used as features to identify the target. Specifically, in the extended CCA, four additional features are used:

$$\mathbf{r}_k = \begin{pmatrix} \mathbf{r}_k(1) \\ \mathbf{r}_k(2) \\ \mathbf{r}_k(3) \\ \mathbf{r}_k(4) \\ \mathbf{r}_k(5) \end{pmatrix} = \begin{pmatrix} \rho(\mathbf{X}^T w_x(\mathbf{X}\mathbf{Y}_k), \mathbf{Y}_k^T w_{\mathbf{Y}_k}(\mathbf{X}\mathbf{Y}_k)) \\ \rho(\mathbf{X}^T w_x(\mathbf{X}\hat{\mathbf{X}}_k), \hat{\mathbf{X}}_k^T w_x(\mathbf{X}\hat{\mathbf{X}}_k)) \\ \rho(\mathbf{X}^T w_x(\mathbf{X}\mathbf{Y}_k), \hat{\mathbf{X}}_k^T w_x(\mathbf{X}\mathbf{Y}_k)) \\ \rho(\mathbf{X}^T w_{\hat{\mathbf{X}}_k}(\hat{\mathbf{X}}_k\mathbf{Y}_k), \hat{\mathbf{X}}_k^T w_{\hat{\mathbf{X}}_k}(\hat{\mathbf{X}}_k\mathbf{Y}_k)) \\ \rho(\hat{\mathbf{X}}_k^T w_{\mathbf{x}}(\mathbf{X}\hat{\mathbf{X}}_k), \hat{\mathbf{X}}_k^T w_{\hat{\mathbf{X}}_k}(\mathbf{X}\hat{\mathbf{X}}_k)) \end{pmatrix} \tag{4}$$

Here, $w_A(AB)$ represents the spatial filter derived from CCA between two multidimensional variables A and B and related to variable A. Then, the sum of these five correlation values is

used as the final feature for target identification:

$$\rho_k = \sum_{i=1}^{5} \mathbf{r}_k(i) \quad , \quad k = 1, 2, \ldots, K \tag{5}$$

Eq (5) also captures the discriminative information from negative correlation coefficients (all except $\mathbf{r}_k(1)$ can be negative). Although the original method uses the sum of the squares of the coefficients along with their signs, in this study, Eq (5) is used due to its superior performance. Finally, the stimulus target is identified by Eq (3).

**2.3.3. TRCA-based method.** TRCA was originally proposed in functional neuroimaging [21] and then used in SSVEP-based BCI to obtain optimized spatial filters to improve SNR of SSVEP response [20]. The method recovers the task-related components (here SSVEP) using a linear, weighted sum of the observed signals (here, multichannel EEG signals):

$$y(t) = \sum_{j=1}^{M} w_j x_j(t) = \mathbf{w}^{\mathrm{T}} \mathbf{x}(t) \tag{6}$$

where $j$ is the index of the channels, $y(t) \in \mathbb{R}$ is the recovered signal, $\mathbf{x}(t) \in \mathbb{R}^M$ is the multi-channel EEG signal, and $\mathbf{w} \in \mathbb{R}^M$ is the optimized spatial filter derived from the TRCA method. This problem can be formulated by maximizing inter-trial covariance [21]. Let $\mathbf{x}^{(h)}(t)$, $y^{(h)}(t)$, and $H$ denote the $h$-th trial of $\mathbf{x}(t)$, the $h$-th trial of $y(t)$, and the number of training trials, respectively. The covariance between the $h_1$-th and $h_2$-th trials of $y(t)$ is defined by:

$$C_{h_1 h_2} = \mathrm{Cov}(y^{(h_1)}(t), y^{(h_2)}(t)) = \sum_{j_1, j_2=1}^{M} w_{j_1} w_{j_2} \mathrm{Cov}(x_{j_1}^{(h_1)}(t), x_{j_2}^{(h_2)}(t)) \tag{7}$$

Then, the sum over all possible combinations of the inter-trial covariance is considered as the objective function:

$$\sum_{\substack{h_1, h_2=1 \\ h_1 \neq h_2}}^{H} C_{h_1 h_2} = \sum_{\substack{h_1, h_2=1 \\ h_1 \neq h_2}}^{H} \sum_{j_1, j_2=1}^{M} w_{j_1} w_{j_2} \mathrm{Cov}(x_{j_1}^{(h_1)}(t), x_{j_2}^{(h_2)}(t)) = \mathbf{w}^{\mathrm{T}} \mathbf{S} \mathbf{w} \tag{8}$$

To limit the weight vector in Eq (8), the variance of y is normalized to one:

$$\mathrm{var}(y(t)) = \sum_{j_1, j_2=1}^{M} w_{j_1} w_{j_2} \mathrm{Cov}(x_{j_1}(t), x_{j_2}(t)) = \mathbf{w}^{\mathrm{T}} \mathbf{Q} \mathbf{w} = 1 \tag{9}$$

The constrained optimization problem then becomes a Rayleigh quotient maximization:

$$\hat{\mathbf{w}} = \arg\max_{\mathbf{w}} \frac{\mathbf{w}^{\mathrm{T}} \mathbf{S} \mathbf{w}}{\mathbf{w}^{\mathrm{T}} \mathbf{Q} \mathbf{w}} \tag{10}$$

The optimal weight vector $\hat{\mathbf{w}}$ is equivalent to the eigenvector corresponding to the largest eigenvalue of the matrix $\mathbf{Q}^{-1}\mathbf{S}$. Then, the following correlation coefficient is computed:

$$\rho_k = \rho(\mathbf{X}^{\mathrm{T}} \mathbf{w}_k, \hat{\mathbf{X}}_k^{T} \mathbf{w}_k) \tag{11}$$

where similar to Subsection 2.3.2, $\mathbf{X}$ and $\hat{\mathbf{X}}_k$ are the single-trial test data and the SSVEP template signal computed by averaging across trials of the $k$-th target, respectively. Also, $\mathbf{w}_k$ is the

spatial filter derived from applying TRCA algorithm on the training data for the $k$-th visual stimulus. In the end, the target can be recognized by the rule provided in Eq (3).

An ensemble TRCA method was proposed in [20] in which the spatial filters derived for different visual stimulus were integrated to construct an ensemble of the spatial filters $\mathbf{W} \in \mathbb{R}^{M \times K}$:

$$\mathbf{W} = [\mathbf{w}_1 \mathbf{w}_2 \ldots \mathbf{w}_K] \tag{12}$$

Since the mixing coefficients from the SSVEP source to the scalp recordings are approximately similar for the utilized frequency range, the $K$ different spatial filters can be considered similar, and this is the reason for the effectiveness of the ensemble TRCA method [20]. In this method, Eq (11) is extended to:

$$\rho_k = \psi(\mathbf{X^T W}, \hat{\mathbf{X}}_k^T \mathbf{W}) \tag{13}$$

where $\psi(A,B)$ indicates the two-dimensional correlation coefficient between $A$ and $B$. Finally, Eq (3) is used for target identification.

## 2.4. Proposed method

The extended CCA method has shortcomings. First, there are numerous ways to project the training data or the test data on the CCA-based spatial filters and compute the correlation between each pair of these projections. Extended CCA uses only five of such correlation coefficients in Eq (4). Also, it is unclear how these five features are selected and the others ignored. Second, there is no ensemble extension for this or any other CCA-based methods. Therefore, these methods cannot compete with ensemble TRCA which has the best performance among the current methods. To mitigate these limitations, in this study, a new method is proposed in which the best CCA-based features are selected. Moreover, to enhance the performance of the method, its ensemble version is also proposed. The structures of the proposed algorithms are illustrated in Fig 1 and their details presented below.

**2.4.1. Basic algorithm.** In the first step, all possible canonical variables (CVs) derived from the CCA-based spatial filters are constructed. In the CCA-based methods, there are three types of data including: 1) the test data $\mathbf{X}$; 2) the template signal $\hat{\mathbf{X}}_k$ derived from averaging across the training blocks of the $k$-th target; and 3) the sinusoidal signals $\mathbf{Y}_k$. By computing CCA between each pair of these three data types, six spatial filters are generated: 1) $\mathbf{W_X}(\mathbf{X}\hat{\mathbf{X}}_k)$; 2) $\mathbf{W}_{\hat{\mathbf{X}}_k}(\mathbf{X}\hat{\mathbf{X}}_k)$; 3) $\mathbf{W_X}(\mathbf{X}\mathbf{Y}_k)$; 4) $\mathbf{W}_{\hat{\mathbf{X}}_k}(\hat{\mathbf{X}}_k \mathbf{Y}_k)$; 5) $\mathbf{W}_{\mathbf{Y}_k}(\mathbf{X}\mathbf{Y}_k)$; and 6) $\mathbf{W}_{\mathbf{Y}_k}(\hat{\mathbf{X}}_k \mathbf{Y}_k)$. Projections of $\mathbf{X}$ and $\hat{\mathbf{X}}_k$ on the first four spatial filters and $\mathbf{Y}_k$ on the 5th and 6th spatial filters generate a total of 10 CVs. These CVs are listed in Table 1.

In the second step, the best correlation features derived from the correlation between each pair of the CVs are found. Since there are 10 CVs, 45 correlation features can be computed ($\begin{pmatrix} 10 \\ 2 \end{pmatrix} = 45$). Fig 2 shows the block diagram of the proposed method for generating the 45 correlation features. Most of these features can be used for target identification. The correlation coefficients between the projections of $\hat{\mathbf{X}}_k$ and the projections of $\mathbf{Y}_k$ (including 8 features) have no capability of detecting SSVEPs even if the test data is used to construct the spatial filters. Also, the correlation between CV9 and CV10 is not useful. Therefore, a combination of the remaining 36 features can be selected for the subject-specific training.

There are a variety of feature selection algorithms in the literature [31, 32]. In this paper, a simple feature selection algorithm called forward selection (FS) [32] is used to find the best set

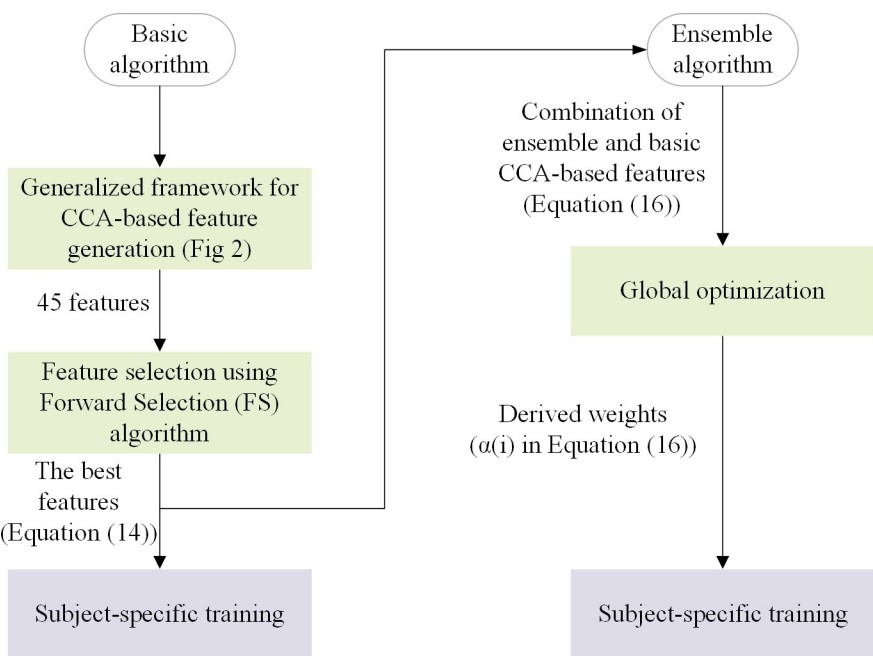

**Fig 1. Structure of the proposed method and its ensemble version.** Green and purple backgrounds represent subject-independent and subject-specific training, respectively.

of correlation features. In this algorithm, the feature which maximizes the average classification accuracy among the 36 features is selected. The classification measure is the same as the one presented in Eq (3). Then, the second feature is selected such that the features selected in the previous and present steps lead to best performance. Similar to Eq (5), the sum of the features is used to combine features for classification. The process of adding features continues until there is no improvement in the average classification accuracy. Finally, the feature set in the last step is considered as the best feature set.

The subject independent training is employed to create the 45 features and apply the FS algorithm on them. After applying the FS algorithm on the seven folds described in Subsection 2.4.3, seven feature sets that contain the best features for each fold are obtained. The interesting point is that in all these feature sets, the maximum performance is provided by the six features that are the same across different folds, although the order in which these features are selected is not the same. Further information regarding features selected in each fold can be found in

**Table 1. Mathematical description of the 10 CVs depicted in Fig 2.**

| Canonical Variable | Formula | Canonical Variable | Formula |
|---|---|---|---|
| $CV_1$ | $\mathbf{X}^T\mathbf{W_X}(\mathbf{X}\hat{\mathbf{X}}_k)$ | $CV_6$ | $\hat{\mathbf{X}}_k^T\mathbf{W_X}(\mathbf{X}\mathbf{Y}_k)$ |
| $CV_2$ | $\hat{\mathbf{X}}_k^T\mathbf{W_X}(\mathbf{X}\hat{\mathbf{X}}_k)$ | $CV_7$ | $\mathbf{X}^T\mathbf{W}_{\hat{\mathbf{X}}_k}(\hat{\mathbf{X}}_k\mathbf{Y}_k)$ |
| $CV_3$ | $\mathbf{X}^T\mathbf{W}_{\hat{\mathbf{X}}_k}(\mathbf{X}\hat{\mathbf{X}}_k)$ | $CV_8$ | $\hat{\mathbf{X}}_k^T\mathbf{W}_{\hat{\mathbf{X}}_k}(\hat{\mathbf{X}}_k\mathbf{Y}_k)$ |
| $CV_4$ | $\hat{\mathbf{X}}_k^T\mathbf{W}_{\hat{\mathbf{X}}_k}(\mathbf{X}\hat{\mathbf{X}}_k)$ | $CV_9$ | $\mathbf{Y}_k^T\mathbf{W}_{\mathbf{Y}_k}(\mathbf{X}\mathbf{Y}_k)$ |
| $CV_5$ | $\mathbf{X}^T\mathbf{W_X}(\mathbf{X}\mathbf{Y}_k)$ | $CV_{10}$ | $\mathbf{Y}_k^T\mathbf{W}_{\mathbf{Y}_k}(\hat{\mathbf{X}}_k\mathbf{Y}_k)$ |

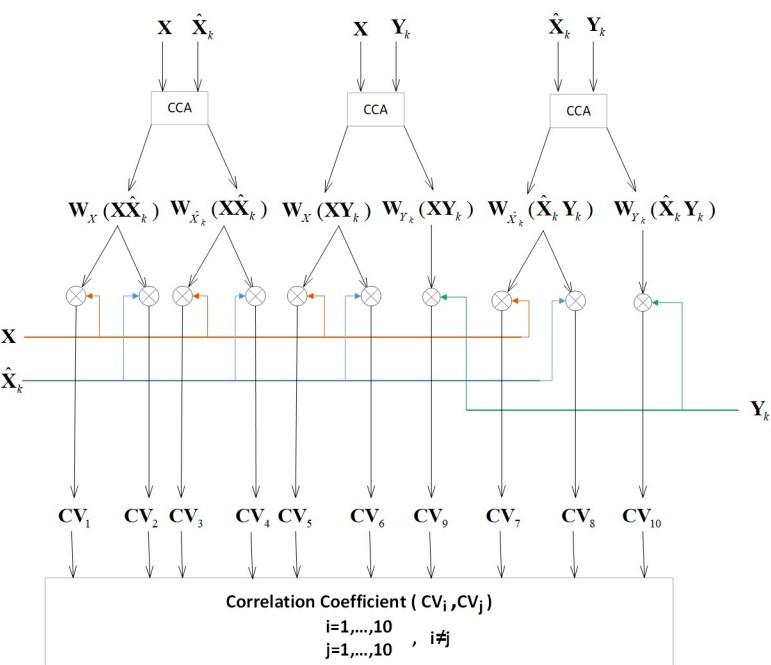

**Fig 2. Block diagram for generating all possible CCA-based correlation features.**

the Supporting information. These six best features are:

$$
\mathbf{r}_k = \begin{pmatrix} \mathbf{r}_k(1) \\ \mathbf{r}_k(2) \\ \mathbf{r}_k(3) \\ \mathbf{r}_k(4) \\ \mathbf{r}_k(5) \\ \mathbf{r}_k(6) \end{pmatrix} = \begin{pmatrix} \rho(\mathbf{X}^\mathbf{T}\mathbf{W}_\mathbf{X}(\mathbf{X}\mathbf{Y}_k), \mathbf{Y}_k^\mathbf{T}\mathbf{W}_{\mathbf{Y}_k}(\mathbf{X}\mathbf{Y}_k)) \\ \rho(\mathbf{X}^\mathbf{T}\mathbf{W}_\mathbf{X}(\mathbf{X}\hat{\mathbf{X}}_k), \hat{\mathbf{X}}_k^\mathbf{T}\mathbf{W}_{\hat{\mathbf{X}}_k}(\mathbf{X}\hat{\mathbf{X}}_k)) \\ \rho(\mathbf{X}^\mathbf{T}\mathbf{W}_\mathbf{X}(\mathbf{X}\hat{\mathbf{X}}_k), \hat{\mathbf{X}}_k^\mathbf{T}\mathbf{W}_\mathbf{X}(\mathbf{X}\hat{\mathbf{X}}_k)) \\ \rho(\mathbf{X}^\mathbf{T}\mathbf{W}_\mathbf{X}(\mathbf{X}\mathbf{Y}_k), \hat{\mathbf{X}}_k^\mathbf{T}\mathbf{W}_\mathbf{X}(\mathbf{X}\mathbf{Y}_k)) \\ \rho(\mathbf{X}^\mathbf{T}\mathbf{W}_{\hat{\mathbf{X}}_k}(\hat{\mathbf{X}}_k\mathbf{Y}_k), \hat{\mathbf{X}}_k^\mathbf{T}\mathbf{W}_{\hat{\mathbf{X}}_k}(\hat{\mathbf{X}}_k\mathbf{Y}_k)) \\ \rho(\mathbf{X}^\mathbf{T}\mathbf{W}_{\hat{\mathbf{X}}_k}(\hat{\mathbf{X}}_k\mathbf{Y}_k), \mathbf{Y}_k^\mathbf{T}\mathbf{W}_{\mathbf{Y}_k}(\hat{\mathbf{X}}_k\mathbf{Y}_k)) \end{pmatrix}
\tag{14}
$$

The coefficients $\mathbf{r}_k(1)$, $\mathbf{r}_k(3)$, $\mathbf{r}_k(4)$, and $\mathbf{r}_k(5)$ are present in both of the extended CCA and the proposed method while the coefficients $\mathbf{r}_k(2)$ and $\mathbf{r}_k(6)$ are exclusively present in our method. These coefficients are used for subject-specific training in the basic algorithm. Similar to Eq (5), the following relation is used to build the final feature for classification:

$$
\rho_k = \sum_{i=1}^{6} \mathbf{r}_k(i) \quad, \quad k = 1, 2, \ldots, K
\tag{15}
$$

**2.4.2. Ensemble algorithm.** Ensemble TRCA showed that an integration of spatial filters derived from calibration data of different classes enhanced performance of the SSVEP BCI [20]. In fact, using both between and within class information in pattern classification methods can boost classifier performance [32]. According to Eq (13), to exploit an ensemble of the spatial filters for a correlation-based feature between two sets, two conditions must be satisfied. First, these two sets should be projected on the same group of spatial filters. Second, the group must contain the spatial filters of all classes. By evaluating these two conditions for the six

features in Eq (14), only $\mathbf{r}_k(3)$, $\mathbf{r}_k(4)$, and $\mathbf{r}_k(5)$ satisfy the first condition and only $\mathbf{r}_k(5)$ satisfies the second condition. Consequently, the six features $\mathbf{r}_k$ in Eq (14) can be converted to the six features $\hat{\mathbf{r}}_k$ in which all features are the same as $\mathbf{r}_k$ except for $\hat{\mathbf{r}}_k(5)$. This feature is constructed using the two-dimensional correlation between two projections on the ensemble of the spatial filters derived from CCA between the template signals $\hat{\mathbf{X}}_k$ and the sinusoidal signals $\mathbf{Y}_k$. Since $\hat{\mathbf{r}}_k(5)$ is the best discriminative feature relative to the other coefficients, a uniform combination of the six coefficients similar to Eq (15) will not be the best solution. To take feature differences into account, a linear weighted sum of the coefficients $\hat{\mathbf{r}}_k(i)$ is proposed:

$$\rho_k = \sum_{i=1}^{6} \alpha(i).\hat{\mathbf{r}}_k(i) \quad , \quad k = 1, 2, \ldots, K \tag{16}$$

The mixing weights $\alpha(i)$ are estimated using the subject independent data (see Subsection 2.4.3). The objective is to maximize the average classification accuracy, computed based on Eqs (3) and (16). Since the objective function is a complex nonlinear function of $\alpha(i)$, the gradient-based optimization methods cannot be easily applied. Considering the limited parameter space of the problem, the metaheuristic optimization methods including the genetic algorithm (GA) or particle swarm optimization (PSO) can be used [33]. We use GA to estimate $\alpha(i)$ coefficients such that the objective function is maximized. GA is implemented using the ga function in MATLAB. For the sake of simplicity and limiting the search space, the coefficients are confined in the [0 1] interval. The estimation process will assign the largest weight ($\alpha(5)$) to $\hat{\mathbf{r}}_k(5)$ due to its highest level of discrimination. Finally, it should be noted that the estimated weights $\alpha(i)$ may be different in different folds.

**2.4.3. Cross-validation.** As mentioned before, both of the subject-independent and the subject-specific trainings are used in the proposed method. Cross-validation is performed on the subjects and the six blocks of a specific subject data for the first and second training techniques, respectively. Further information about cross-validation techniques is presented below.

**Subject-independent training:** The parts related to this training technique are shown in green in Fig 1. In this approach, the K-fold (K = 7) approach is used and the data of 30 subjects is utilized to obtain the best hyperparameters for the remaining 5 subjects. Then, the obtained hyperparameters are used to create the subject-specific models. Specifically, in the basic algorithm, for each fold, 45 CCA-based features are constructed for the 30 subjects and then, the features that maximize the average recognition accuracy for the mentioned subjects are selected (Subsection 2.4.1). Finally, the subject-specific models are created for the remaining 5 subjects using the selected features. Similarly, in the ensemble algorithm, the weights (Subsection 2.4.2) that maximize the average accuracy for the 30 subjects of the corresponding fold are used to build the subject-specific models of the remaining subjects. Therefore, the selected features in the basic algorithm and the weights $\alpha(i)$ in the ensemble algorithm are considered as the hyperparameters.

**Subject-specific training:** In both of the basic and ensemble algorithms, the subject-specific models are built using the hyperparameters derived from the other subjects' data. For each subject, the leave-one-out technique is used on the six blocks. In other words, the data samples from five of the six blocks are used as the training data to construct a reference signal for each target while the left-out (sixth) block is used for validation. This procedure is repeated six times such that every block is considered as validation data once. Finally, the average recognition accuracy across these six blocks are computed. It is worthwhile to note that the classification accuracies reported in the Result Section are from this type of training.

## 2.5. Filter bank analysis

Higher harmonics of the SSVEP stimulus frequency contain useful information which can improve the recognition accuracy. To extract this information, filter bank analysis has been proposed as a practical solution in which a signal is decomposed to multiple frequency sub-bands [29, 34]. Filter bank analysis can reduce the detection error due to the background EEG activities. X. Chen, et al. [29] applied the filter bank technique to the SSVEP-based BCI, enhancing the performance of the standard CCA method significantly. This technique is applied to all methods presented here and its effect is reported. To design the filter bank, a procedure similar to [12, 29] is utilized. In this method, the EEG data is decomposed into N sub-bands using the N band-pass filters and a feature extraction algorithm is applied to each sub-band separately. The lower and upper cut-off frequencies of the $n$-th sub-band are set to n×8 Hz and 70 Hz, respectively. The zero-phase Chebyshev Type II IIR band-pass filter is used to extract every sub-band signals. The features computed from the sub-bands are combined as follows:

$$\tilde{\rho}_k = \sum_{n=1}^{N} w_{SB}(n)\rho_k^{(n)} \tag{17}$$

where $\rho_k^{(n)}$, $\tilde{\rho}_k$, and $w_{SB}(n)$ are the feature value for the $n$-th sub-band and the $k$-th target, the final feature for classification, and the weights for the sub-band components, respectively. Based on the previous studies, when the response frequency increased, the SNR of SSVEP decreased [29]. Therefore, the sub-band weights are determined using:

$$w_{SB}(n) = n^{-a} + b, \quad n \in [1\ N] \tag{18}$$

Following [12], $a$ and $b$ are set to 1 and 0, respectively. As mentioned before, the target is selected by Eq (3) and substituting $\rho_k$ with $\tilde{\rho}_k$.

## 3. Results

Classification accuracy and ITR were used as the evaluation metrics to compare the performance of the methods. These two metrics were calculated with various data lengths from 0.2 s to 1 s with a step of 0.1 s. The 0.5 s gaze shifting duration was considered to compute the simulated ITR in the offline analysis. Also, the number of harmonics in Eq (1) was set to 3. Fig 3 shows the average accuracies and ITRs across subjects for three basic methods at different time windows, with and without the filter bank. For the filter bank, the number of sub-bands was set to 4. In all possible cases, TRCA showed a superior performance over the other methods for the time windows shorter than 0.3 s. For the 0.3 s time window, the one-way repeated measures analysis of variance (ANOVA) showed no significant difference between the accuracy (F(2,68) = 1.35, p = 0.26) and ITR (F(2,68) = 1.09, p = 0.33) of the three methods without the filter bank. When filter bank was applied in the 0.3 s time window, ANOVA revealed significant difference in the accuracy (F(2,68) = 17.79, p<0.001) and ITR (F(2,68) = 18.45, p<0.001) of the three methods. The post-hoc paired t-tests showed that there was no significant difference in accuracy (p = 0.67) and ITR (p = 0.62) between the TRCA method and the proposed method while both methods outperformed the extended CCA method (p<0.001). For time windows greater than 0.3 s, ANOVA indicated significant difference (p<0.01) between the three methods in all conditions. Post-hoc paired t-tests confirmed superior performance of the proposed method relative to TRCA and extended CCA (p<0.01). In Fig 3B, the time windows corresponding to the highest ITR are different for each method (extended CCA: 0.8 s; TRCA: 0.8 s; the proposed method: 0.7 s) while in Fig 3D, all methods reached their highest ITR in 0.7 s.

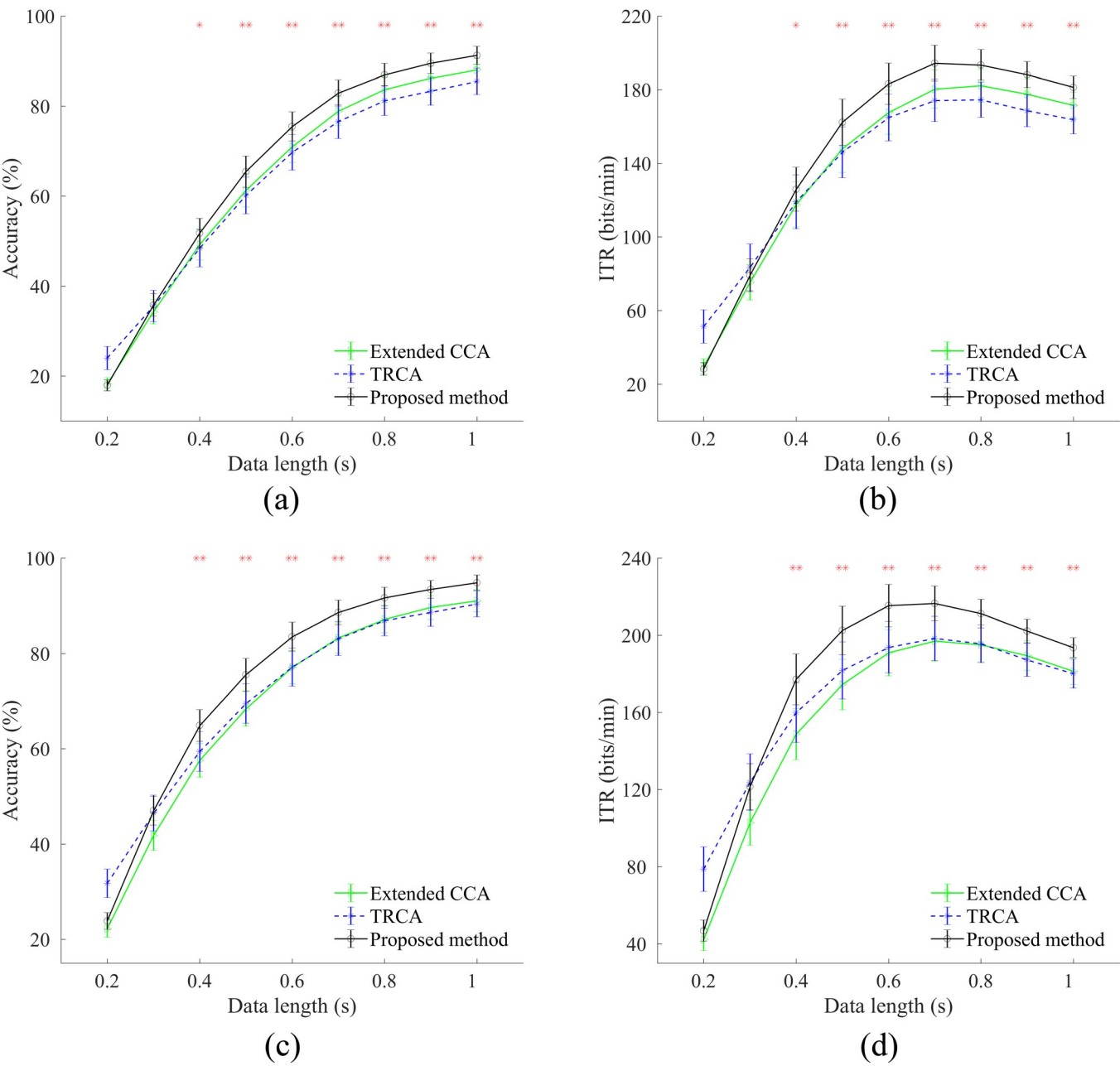

**Fig 3. Average accuracies, (a) and (c), and ITRs, (b) and (d), across subjects for three basic methods at different time windows.** Results in the first and second rows are derived without and with the filter bank, respectively. Number of sub-bands is set to 4. Asterisks represent significant difference between the three methods, using ANOVA at time windows greater than 0.3 ($^{*}$p<0.01, $^{**}$p<0.001). Error bars show standard errors.

The ensemble version of the proposed method is compared with the ensemble TRCA method in Fig 4. To estimate the weights ($\alpha(i)$) in Eq (16) using the procedure described in Subsection 2.4.2, the time window was set to 0.5 s. Similar to the basic methods, the ensemble TRCA method performed better than the proposed ensemble method in all cases when the data length was less than 0.3 s. For 0.3 s, paired t-tests showed no significant difference between the two methods, with and without filter bank (Fig 4A: p = 0.62; Fig 4B: p = 0.50; Fig 4C: p = 0.12; Fig 4D: p = 0.35). For the data lengths greater than 0.3 s, the proposed ensemble

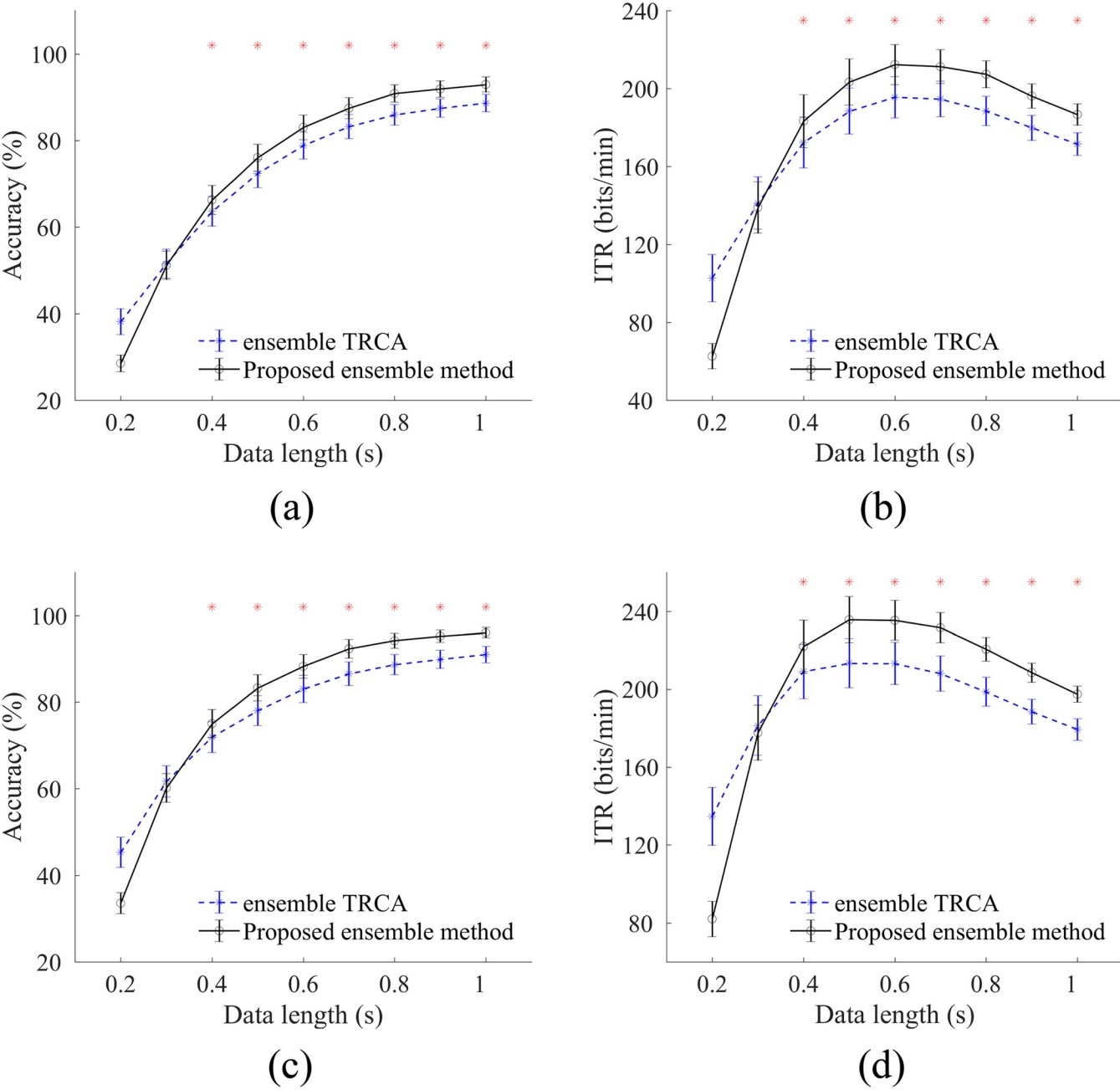

**Fig 4. Average accuracies, (a) and (c), and ITRs, (b) and (d), across subjects for ensemble TRCA and ensemble version of the proposed method at different time windows.** Results in the first and second rows are derived without and with the filter bank, respectively. Number of sub-bands is set to 4. Asterisks represent significant difference between the two methods by paired t-tests at time windows greater than 0.3 (*p<0.001). Error bars show standard errors.

method led to significantly (p<0.001) higher accuracy and ITR than the ensemble TRCA method for both cases. Both methods reached their highest ITRs at 0.6 s in Fig 4B and 0.5 s in Fig 4D.

The performance of the training methods depends on the number of sub-bands, electrodes, and training blocks. Therefore, the effects of varying these parameters on the classification accuracy for all cases including the basic and ensemble TRCA, and the basic and ensemble

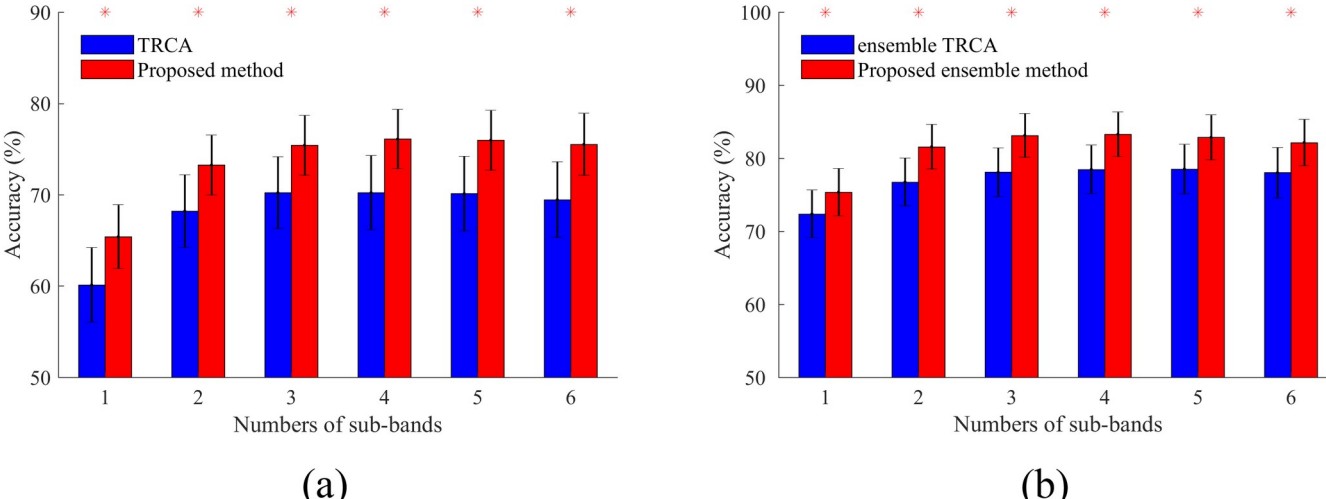

**Fig 5. Average accuracies across subjects for different number of sub-bands.** (a) Basic TRCA and the proposed method; and (b) ensemble TRCA and the proposed ensemble method. Asterisks show significant differences between the two methods by paired t-tests (*p<0.001). Error bars show standard errors.

version of the proposed method are investigated in Figs 5 and 6. Time window was set at 0.5 s to perform the analysis. In Fig 5, the number of the training blocks and the electrodes were fixed at 5 and 9 and the effect of the number of sub-bands was explored. The proposed method represents significantly (p<0.001) higher classification accuracies than TRCA in all cases. For both of the basic and the ensemble versions of the two methods, the highest accuracy is achieved by 4 sub-bands. According to this fact, the number of sub-bands was fixed at 4 and the variations of the average accuracies corresponding to different numbers of the electrodes and the training blocks were examined in Fig 6. The results illustrate that for both of the basic and ensemble cases, the proposed method outperforms TRCA, especially for low numbers of the training blocks and the electrodes (p<0.001). Furthermore, TRCA needs at least two training blocks to obtain optimal spatial filters while the proposed method can deliver an acceptable performance even with a single training block (see Fig 6B and 6D). This characteristic can be one of the major advantages of our method compared with TRCA. Typically, in SSVEP BCI, it is necessary to collect the training data at the beginning of each session which could be time-consuming; our method reduces the training time considerably.

## 4. Discussion

Classification accuracy and ITR are the most important factors for practical development of SSVEP-based BCI spellers and thus must be improved as much as possible. In this study, an ensemble CCA-based training method was proposed for the first time, which improved the performance of the extended CCA and TRCA methods. The proposed method outperformed extended CCA in all conditions. Furthermore, it outperformed TRCA in terms of both accuracy and ITR for data lengths greater than 0.3 s. The lower performance of our method for short time durations could be related to inaccurate estimation of the spatial filters by the CCA algorithm from a small number of samples. However, when the data length increases, on one hand, the spatial filters are estimated more accurately and on the other hand, the combination of various coefficients which exploit CCA-based spatial filters improve the performance of the proposed method compared with TRCA.

In practical applications, for majority of the subjects, the maximum speed (highest ITR) is reached at time windows greater than 0.3 s, justifying the application of the proposed method

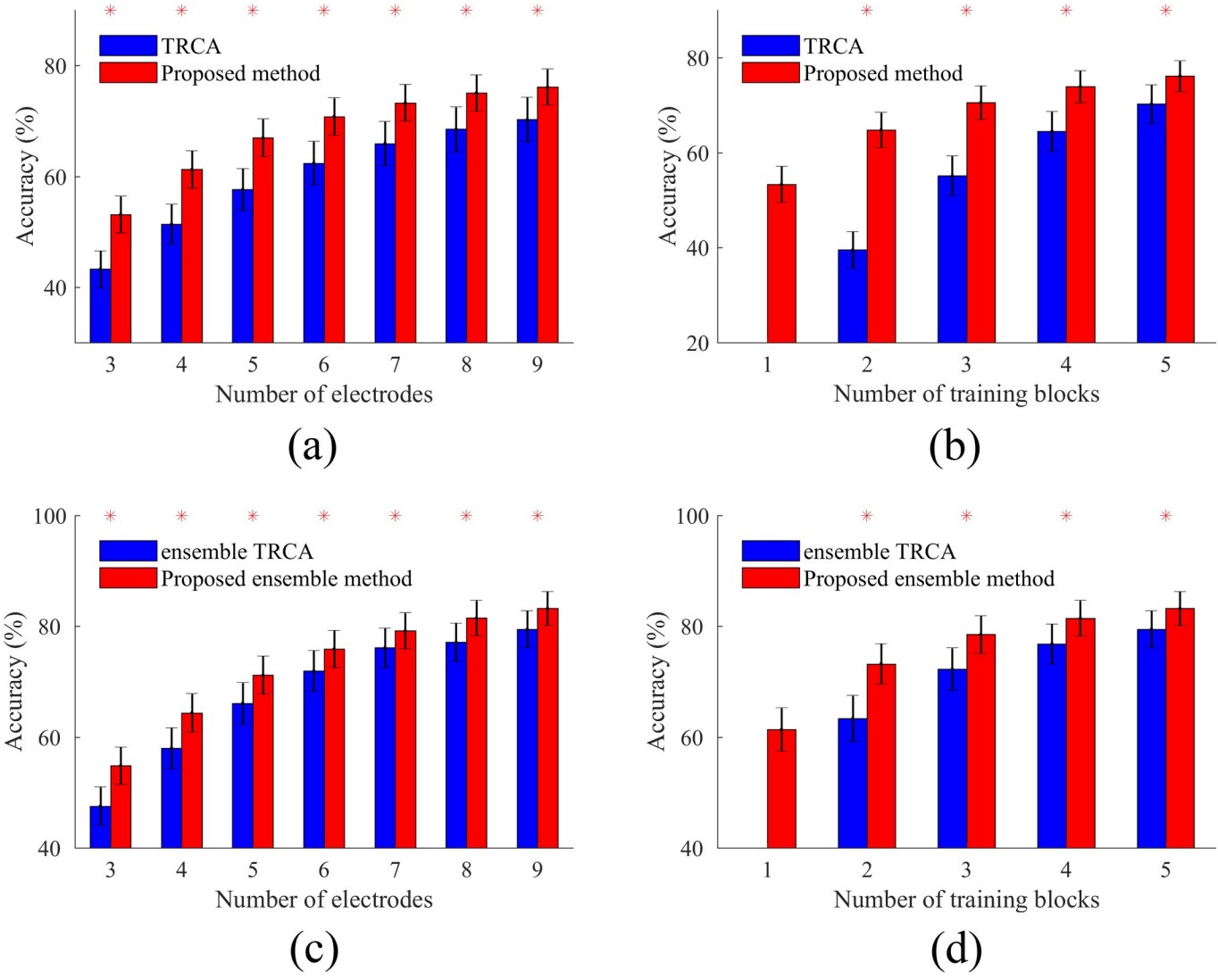

**Fig 6. Average accuracies across subjects obtained by different number of electrodes, (a) and (c), and training blocks, (b) and (d).** The first row compares two basic methods and the second row compares two ensemble methods. Asterisks show significant differences between the two methods by paired t-tests ($^*$p<0.001). Error bars show standard errors.

for such subjects. All in all, only when the numbers of the blocks and the electrodes are large and the subject reaches his/her highest ITR in 0.3 s or less, the TRCA method is preferable to the proposed method. Otherwise, the proposed method is recommended. Also, in this paper, due to the limited number of training blocks per subject, the subject-independent training technique was used to find the best CCA-based features and estimate the mixing weights in Eq (16). For a new subject, Eqs (14), (15) and (16), and one set of weights $\alpha(i)$ are sufficient for target detection.

For further investigation of the performance of the proposed method relative to TRCA, feature values can be compared for the two methods. Since the scales of the final features obtained by the two methods are different, feature vectors derived from each trial are linearly normalized into [-1, +1] and then compared. Fig 7A and 7B represent normalized feature values for a sample frequency derived from two basic and two ensemble methods, respectively. The number of sub-bands, electrodes, and training blocks were 4, 9, and 5, respectively. A short data

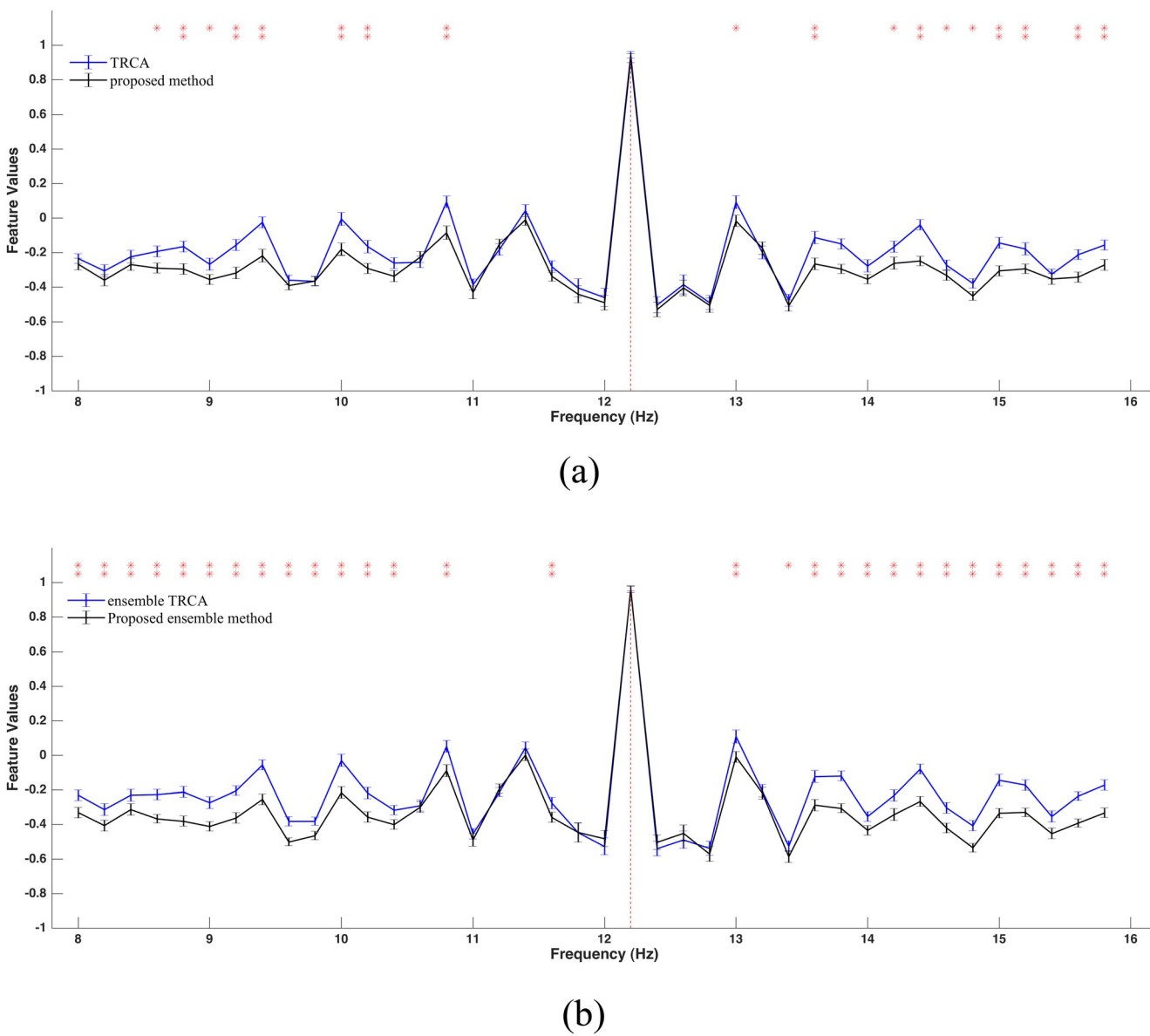

**Fig 7. An example of normalized feature values, averaged across subjects and blocks, obtained by: (a) two basic methods; and (b) two ensemble methods.** Red vertical line indicates true frequency. Data length is 0.6 s. Asterisks represent a significant difference between the two methods by paired t-tests ($^*$p<0.01, $^{**}$p<0.001). Error bars show standard errors.

length (0.6 s) was selected to carry out comparisons. In both figures, the feature values of the two methods decline with a similar trend in the neighborhood of the true frequency. However, as we move away from the true frequency, feature values of the proposed method become significantly (p<0.001) lower than those of the TRCA method. Therefore, the probability of a false detection in our method is lower than that of TRCA, leading to its superiority over TRCA.

There are several parameters in this paper which can be further optimized for each method (or subject) separately, including the filter bank design, the stimulus design, and the electrode setting. As a representative example, consider different possible sets of $n$ ($n<9$) electrodes which can be selected from the nine electrodes introduced in Subsection 2.2. For an $n$, the

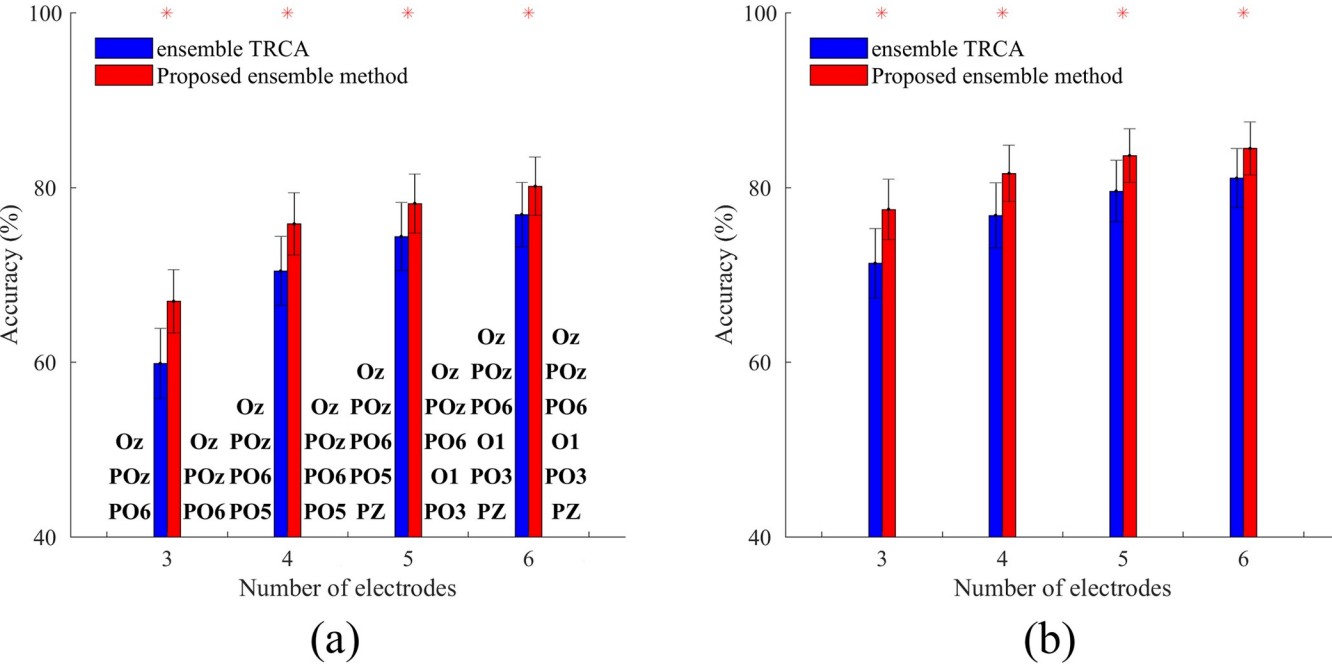

**Fig 8. Relation between the maximum achievable accuracy and the layout of the electrodes.** (a) the best layout of the electrodes per method, derived from a grid search for all subjects and the corresponding average accuracies; and (b) the potential average accuracies across the subjects after selecting the best layout of the electrodes per subject. In both figures, the data length is 0.5 s. Asterisks represent a significant difference between the two methods by paired t-tests (*p<0.001). Error bars show standard errors.

optimal electrode layout per method can be found by a grid search, i.e., by calculating average accuracies across the subjects for each layout and selecting the layout with the highest accuracy. This analysis is done on the benchmark dataset with three to six electrodes for the proposed ensemble method and the ensemble TRCA method. Then, the best layout per method along with the corresponding accuracies are shown in Fig 8A. This figure shows that by selecting a suitable subset of four or five electrodes, acceptable accuracies, comparable with those obtained by nine electrodes, can be achieved. It also illustrates that if we consider a local area (i.e., visual area), the best layout obtained by a grid search is almost independent of the spatial filter-based target identification method used.

Another approach for optimizing the electrode setting is the channel selection in an unsupervised manner [35]. The maximum achievable accuracy per subject derived from a grid search can be used as a reference to compare the performance of the channel selection algorithms in the future studies. For example, Fig 8B shows average accuracies after selecting the best electrodes per subject. This figure reveals the great potential of an effective channel selection algorithm to enhance the performance of the methods. Superior performance of the proposed method compared with TRCA is illustrated in both Fig 8A and 8B.

In this study, a method was proposed which uses both of the subject-specific and the subject-independent training techniques. Since collecting the training data is time-consuming and may be exhausting for some subjects, the transfer learning methods have been proposed which use the training data of the other subjects [24] or different sessions of the same subject [36]. Furthermore, using the benchmark dataset containing a large number of subjects [25], various training-free algorithms can be devised and evaluated in the future studies to improve effectiveness of such methods. Since the optimal data length for various trials can be different, an adaptive selection of the window length using a dynamic stopping criterion can be a

solution for the BCI users [37–38]. Besides, the combination of SSVEP and other modalities, e.g., the eye-tracking systems [39], can improve the performance compared with using two single-modality methods. However, the efficiency of the hybrid methods over the single-modality methods needs to be investigated.

The advantages of our approach relative to the TRCA and extended CCA methods for target detection in SSVEP-based BCI can be summarized as the following.

- Our method integrates subject-specific models with subject-independent information and enhances the BCI performance.

- The classification accuracy and information transfer rate (ITR) of our method are significantly higher than those of the extended CCA in all conditions and those of TRCA in time windows larger than 0.3 s.

- Our method can be easily implemented in online applications of BCI and realize a high-speed SSVEP based speller.

- Our method outperforms TRCA when the number of the training blocks and the number of the electrodes are small. Also, for subject-specific training, TRCA needs at least two training blocks while our method works with a single training block. This facilitates the development and application of the BCI systems.

- A problem with the SSVEP-based BCI spellers is false detection, due to interference from the nearest neighbors of the target frequency. The likelihood of this error for our method is lower than that of the TRCA method.

## 5. Conclusion

This study proposed a framework to improve traditional CCA-based training methods by finding the best hyperparameters for each subject using other subjects' training data. These hyperparameters were used to construct the basic and ensemble versions of the proposed method. The offline analysis based on a benchmark dataset was performed and the proposed method was compared with the extended CCA and TRCA methods. Our method showed significantly higher performance than extended CCA in all conditions and TRCA in time windows greater than 0.3 s. All three methods can be implemented in online BCI applications to realize a high-speed SSVEP-based speller.

## Supporting information

**S1 Appendix. Feature selection results.**
(DOCX)

## Acknowledgments

The authors would like to thank the authors of [25] for providing the benchmark dataset freely.

## Author Contributions

**Conceptualization:** Mohammad Hadi Mehdizavareh, Sobhan Hemati.

**Data curation:** Mohammad Hadi Mehdizavareh.

**Formal analysis:** Mohammad Hadi Mehdizavareh, Sobhan Hemati.

**Investigation:** Hamid Soltanian-Zadeh.

**Methodology:** Mohammad Hadi Mehdizavareh.

**Software:** Mohammad Hadi Mehdizavareh.

**Supervision:** Hamid Soltanian-Zadeh.

**Validation:** Sobhan Hemati, Hamid Soltanian-Zadeh.

**Visualization:** Mohammad Hadi Mehdizavareh.

**Writing – original draft:** Mohammad Hadi Mehdizavareh, Sobhan Hemati.

**Writing – review & editing:** Hamid Soltanian-Zadeh.

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
