## [Decision Letter · Decision Letter 0]

25 Sep 2019

PONE-D-19-20684

Enhancing performance of subject-specific models via subject-independent information for SSVEP-based BCIs

PLOS ONE

Dear Prof. Soltanian-Zadeh,

Thank you for submitting your manuscript to PLOS ONE. After careful consideration, we feel that it has merit but does not fully meet PLOS ONE’s publication criteria as it currently stands. Therefore, we invite you to submit a revised version of the manuscript that addresses the points raised during the review process.

We would appreciate receiving your revised manuscript by Nov 09 2019 11:59PM. To enhance the reproducibility of your results, we recommend that if applicable you deposit your laboratory protocols in protocols.io, where a protocol can be assigned its own identifier (DOI) such that it can be cited independently in the future. For instructions see: http://journals.plos.org/plosone/s/submission-guidelines#loc-laboratory-protocols

We look forward to receiving your revised manuscript.

Kind regards,

Xiang Gao, Ph.D

Academic Editor

PLOS ONE

Journal Requirements:

Additional Editor Comments (if provided):

I am pleased to inform you that your manuscript is acceptable forvpublication in Plos one pending minor revision.

Below is a link to the decision and reviewers' comments regarding your submission.

Please revise your manuscript according to the reviewers' suggestions and provide a point-by-point response to the reviews.

Reviewers' comments:

Reviewer's Responses to Questions

**Comments to the Author**

1. Is the manuscript technically sound, and do the data support the conclusions?

Reviewer #1: Yes

Reviewer #2: Yes

2. Has the statistical analysis been performed appropriately and rigorously? 

Reviewer #1: Yes

Reviewer #2: Yes

3. Have the authors made all data underlying the findings in their manuscript fully available?

Reviewer #1: Yes

Reviewer #2: Yes

4. Is the manuscript presented in an intelligible fashion and written in standard English?

Reviewer #1: Yes

Reviewer #2: Yes

5. Review Comments to the Author

Reviewer #1: Authors have proposed a framework to improve traditional CCA-based training methods by finding the best hyperparameters for each subject using other subjects’ training data. It showed significantly higher performance than extended CCA in all conditions and TRCA in time windows greater than 0.3s. This proposed method is innovative and significant, which is helpful for disabled individuals. However, there are still some questions.

1. Authors used publicly available 35-subject SSVEP benchmark dataset to evaluate the proposed method, and data of 30 subjects is utilized to obtain the best hyperparameters for the remaining 5 subjects for cross-validation, is the number of subjects enough for comparation?

2. The figures are arranged in all of the paper including methods, results and discussion, it seems a little messy. In addition, authors may consider expanding the heading fonts of abscissa and ordinate, such as Figure3 and Figure4.

3. Authors may pay attention to the format of references, such as reference 18 and 39.

Reviewer #2: 1. The figures are not clear. Please provide high resolution figures.

2. The paper is written in poor English. It requires extensive language revision, possibly by a native English speaker.

3. Please describe the broad value and interest of your new approach.

6. PLOS authors have the option to publish the peer review history of their article (what does this mean?). If published, this will include your full peer review and any attached files.

Reviewer #1: No

Reviewer #2: No

---

## [Author Response · Author response to Decision Letter 0]

29 Oct 2019

Point-by-Point Responses to the Reviewers’ Comments

We thank the reviewers for their valuable comments and suggestions, based on which we have revised the manuscript. We have kept the track of the changes in the revised manuscript and provided our point-by-point responses to the comments below.

Reviewer 1 

Comment 1: Authors used publicly available 35-subject SSVEP benchmark dataset to evaluate the proposed method, and data of 30 subjects is utilized to obtain the best hyperparameters for the remaining 5 subjects for cross-validation, is the number of subjects enough for comparation?

Response: The mentioned process was repeated 7 times for the 7 folds. For each fold, the data of 30 subjects were used to find the best hyperparameters. Then, these hyperparameters were used for the remaining 5 subjects. Therefore, we used the data of all 35 subjects (7 times 5) in the evaluation process. We concatenated the results of all folds (the output became a vector with a length of 35) and averaged across the subjects. Therefore, the reported results are the average performance across all subjects. The number of subjects is sufficient for the comparison, similar to previous works that used this dataset.

Comment 2: The figures are arranged in all of the paper including methods, results and discussion, it seems a little messy. In addition, authors may consider expanding the heading fonts of abscissa and ordinate, such as Figure 3 and Figure 4.

Response: Majority of the figures (6 out of 8) are presented in the Methods and Results Sections to explain the proposed methods and the results obtained from their applications. We used two figures in the Discussion Section to illustrate and compare additional features of the methods. We have applied the suggested changes to Figures 3 and 4. 

Comment 3: Authors may pay attention to the format of references, such as reference 18 and 39.

Response: We have corrected the format of references including references 18 and 39 in the revised manuscript. 

Reviewer 2 

Comment 1: The figures are not clear. Please provide high resolution figures.

Response: We think the problem is with the pdf file generated by the online submission website. As acknowledged by PLOS ONE, the quality of the images in the generated pdf may be low. However, PLOS ONE provides a link for dowloading the high-quality version of each image. Our submitted figures had a resolution of 300 dpi and the dimensions were corrected using https://pacev2.apexcovantage.com as required by the journal. To be on the safe side, we double-checked our submitted images on the PLOS ONE website and confirmed their high quality. Please let us know if there is any additional requirements (e.g., a resolution higher than 300 dpi or a dimension larger than the current version). Also, the .mat version of the images is available if needed. 

Comment 2: The paper is written in poor English. It requires extensive language revision, possibly by a native English speaker.

Response: We have extensively revised the manuscript to improve its English.

Comment 3: Please describe the broad value and interest of your new approach.

Response: The advantages of our approach relative to the TRCA and extended CCA methods (two state-of-the-art methods) for target detection in SSVEP-based BCI are the following, which are also presented in the revised manuscript.

• Our method integrates subject-specific models with subject-independent information and enhances the BCI performance.

• The classification accuracy and information transfer rate (ITR) of our method are significantly higher than those of the extended CCA in all conditions and those of TRCA in time windows larger than 0.3 s. 

• Our method can be easily implemented in online applications of BCI and realize a high-speed SSVEP based speller.

• Our method outperforms TRCA when the number of the training blocks and the number of the electrodes are small. Also, for subject-specific training, TRCA needs at least two training blocks while our method works with a single training block. This facilitates the development and application of the BCI systems. 

• A problem with the SSVEP-based BCI spellers is false detection, due to interference from the nearest neighbors of the target frequency. The likelihood of this error for our method is lower than that of the TRCA method.

---

## [Decision Letter · Decision Letter 1]

19 Nov 2019

Enhancing performance of subject-specific models via subject-independent information for SSVEP-based BCIs

PONE-D-19-20684R1

Dear Dr. Hamid Soltanian-Zadeh,

We are pleased to inform you that your manuscript has been judged scientifically suitable for publication and will be formally accepted for publication once it complies with all outstanding technical requirements.

With kind regards,

Xiang Gao, Ph.D

Academic Editor

PLOS ONE

Additional Editor Comments (optional):

I am very pleased to inform you that your revised manuscript "Enhancing performance of subject-specific models via subject-independent information for SSVEP-based BCIs" has been accepted for publication in the Plos one.

Reviewers' comments:

Reviewer's Responses to Questions

**Comments to the Author**

1. If the authors have adequately addressed your comments raised in a previous round of review and you feel that this manuscript is now acceptable for publication, you may indicate that here to bypass the “Comments to the Author” section, enter your conflict of interest statement in the “Confidential to Editor” section, and submit your "Accept" recommendation.

Reviewer #1: All comments have been addressed

Reviewer #2: (No Response)

2. Is the manuscript technically sound, and do the data support the conclusions?

Reviewer #1: Yes

Reviewer #2: (No Response)

3. Has the statistical analysis been performed appropriately and rigorously? 

Reviewer #1: Yes

Reviewer #2: (No Response)

4. Have the authors made all data underlying the findings in their manuscript fully available?

Reviewer #1: Yes

Reviewer #2: (No Response)

5. Is the manuscript presented in an intelligible fashion and written in standard English?

Reviewer #1: Yes

Reviewer #2: (No Response)

6. Review Comments to the Author

Reviewer #1: Authors had revised as my comments, including the format and figures, I agree to accept this manuscript

Reviewer #2: (No Response)

7. PLOS authors have the option to publish the peer review history of their article (what does this mean?). If published, this will include your full peer review and any attached files.

Reviewer #1: No

Reviewer #2: No

---

## [Editor Report · Acceptance letter]

3 Jan 2020

PONE-D-19-20684R1 

Enhancing performance of subject-specific models via subject-independent information for SSVEP-based BCIs 

Dear Dr. Soltanian-Zadeh:

I am pleased to inform you that your manuscript has been deemed suitable for publication in PLOS ONE. Congratulations! Your manuscript is now with our production department. 

With kind regards,

on behalf of

Dr. Xiang Gao 

Academic Editor

PLOS ONE